# There Are Predictors of Eating Disorders among Internet Use Characteristics—A Cross-Sectional Study on the Relationship between Problematic Internet Use and Eating Disorders

**DOI:** 10.3390/ijerph181910269

**Published:** 2021-09-29

**Authors:** Marta Kożybska, Iwona Radlińska, Aleksandra Czerw, Grażyna Dykowska, Beata Karakiewicz

**Affiliations:** 1Subdepartment of Medical Law, Department of Social Medicine, Pomeranian Medical University, ul. Żołnierska 48, 71-210 Szczecin, Poland; marta.kozybska@pum.edu.pl; 2Department of Health Economics and Medical Law, Medical University of Warsaw, ul. Żwirki i Wigury 81, 02-091 Warsaw, Poland; aleksandra.czerw@wum.edu.pl (A.C.); grazyna.dykowska@wum.edu.pl (G.D.); 3Subdepartment of Social Medicine and Public Health, Department of Social Medicine, Pomeranian Medical University, ul. Żołnierska 48, 71-210 Szczecin, Poland; beata.karakiewicz@pum.edu.pl

**Keywords:** Internet addiction, problematic Internet use, eating disorders, students, Poland

## Abstract

The aims of this cross-sectional study were: (i) to establish the prevalence of problematic Internet use (PIU) and eating disorders (EDs) among Polish students; (ii) to investigate potential correlations between the two phenomena; and (iii) to identify predictors of eating disorders among socio-demographic and Internet use characteristics in this population. To this end, a total of 1008 Polish students aged 18–40, completed the Problematic Internet Use Test (TPIU22), the Eating Attitudes Test (EAT-26) and a self-designed Socio-demographic and Internet Use Survey. Men received more PIU scores (*p* < 0.001), while women received more EAT-26 scores (*p* < 0.05) with a significant correlation observed between those variables (rho = 0.212; *p* < 0.001). The strongest predictors of EDs were as follows: preoccupation with the Internet, neglect of sleep in favor of Internet use, alleviation of negative feelings while online, higher mean number of hours spent online on weekends for academic and work-related purposes, extracurricular activity, lower height and higher BMI. An association has been demonstrated between problematic internet use and eating disorders. Somewhat surprisingly, our results suggest that people at risk of EDs use the Internet primarily to fulfill their routine duties. Nevertheless, further research is needed to establish the causality of EDs and PIU.

## 1. Introduction

Observations in recent years have made it clearer that the Internet is not only a helpful tool facilitating learning, work and entertainment, but also a source of threat and negative functional outcomes, assuming a form of addiction. In the literature, this very phenomenon is referred to in various ways as “compulsive Internet use”, “Internet related problems”, “pathological Internet use”, “Internet addiction”, or “excessive Internet use,” to name but a few. A quite compelling term emphasizing its negative consequences is “problematic Internet use” (PIU), understood as one that results in psychological, social, academic and occupational difficulties [1]. Evidence suggests that heavy use over time alone should not be the only criterion for measuring PIU [2], even though longer on-line activity has been demonstrated to serve as its strong predictor [3,4,5,6,7]. Research to date also implicates the purpose of Internet use in the development of PIU. Indeed, on-line activity is more frequently associated with entertainment rather than study or work-related purposes [5,8,9,10]. Young men are particularly at risk of developing PIU [4,5,11,12,13,14]. Some studies suggest that PIU severity may be linked to the selected university major [15,16], although others yield contradictory results [8]. The relationship between self-reported financial status and PIU remains unclear [4,11,17].

Researchers distinguish two types of Internet addiction: generalized [18] and specific [19], the former involving general use of the Internet, while the latter relating to addiction to particular activities performed on the Internet, e.g., excessive gaming, sexual preoccupations, and email/text messaging [19]. Factors associated with the development of specific Internet-use disorders are divided into two groups: person’s core characteristics, including personality traits such as high impulsivity, low self-esteem, low conscientiousness, high shyness, high neuroticism, low self-directedness, low self-efficacy, vulnerability to stress, procrastination tendencies; and psychopathology, including depression, social anxiety, ADHD (interaction of the person–affect–cognition–execution model) [20].

Excessive preoccupation with the internet affects various aspects of life, including eating behaviors. Internet addiction has been associated with both excessive and too low body weight [21,22], junk food consumption [17], and a generally unhealthy lifestyle [21]. The relationship between PIU and BMI remains unclear [23,24,25,26]. However, food addiction has been suggested to be a predictor of Internet addiction, and the two phenomena were also described in terms of overlapping addiction [23].

Of particular note are the psychological aspects of the relationship between PIU and EDs. For example, individuals with elevated levels of body image concerns and EDs may be more at risk of developing PIU, and PIU may be related to body image avoidance [27]. PIU has even been postulated to influence ED risk more than exercise dependence [28]. To the best of our knowledge, researchers have been pondering the relationship between PIU and EDs since the year 2009 [25]. To date, however, due to the relative paucity of evidence, the answer to the question of whether there is one still remains unclear [29]. What ED and PIU may in fact have in common could be the socio-cultural context. Evidence has been demonstrated for a significant effect of social and cultural factors on the development of EDs [30,31].

The Internet, in turn, is an extraordinary medium not only of culture, but also of social interaction, especially during the SARS-CoV-2 pandemic. To this day, researchers have been struggling with establishing the causality of the PIU’s relationship with EDs. Perhaps those involved in excessive Internet use succumb to body images promoted in the mass media and, therefore, feel dissatisfied with their own appearance, which may subsequently lead to EDs [32]. Alternatively, those dissatisfied with their physical appearance, at risk of developing ED, treat the Internet as an escape from direct, real-life contact, considering the virtual world a good, safe venue of social interaction. This remains in line with the fact that low self-esteem is among the risk factors of Internet addiction [23]. Especially disturbing are reports from Poland, suggesting that girls who use laxatives and diets manifest a greater degree of Internet addiction than their non-dieting counterparts. Not only that, these girls are more likely to believe that the Internet is the only space for them to talk about their real feelings, voice opinions, and discuss important matters [33]. Unfortunately, there are various forums and websites designed to associate people with EDs and promote EDs as a lifestyle choice [33].

So far, studies on the relationship between PIU and EDs have been conducted mainly among students, in Turkey, Spain, Taiwan, Egypt, Colombia, USA, China, on samples ranging from 314 to 2780 participants [29]. Quite interestingly, this issue has not been extensively investigated by European researchers [29]. It, therefore, suggests that empirical research in this area is still in its infancy [29]. Very little is known about the relationship between PIU and EDs and about predictors of EDs among computer use characteristics. With an aim to fill this gap, we conducted a cross-sectional study among Polish students, formulating the following objectives:

To establish the prevalence of PIU and ED among Polish students, taking into account sex of the participants. Based on previous literature reports, we expected greater severity of PIU in men relative to women [4,5,11,12,13,14]. We also expected greater likelihood of ED-related risk in women than in men (due to their reported greater preoccupation with the body image [34]).

To investigate a potential correlation between PIU and Eds, we hypothesized that PIU scores would be higher in individuals with higher ED scores (as indicated in previous reports [29]).

To identify which sociodemographic variables and characteristics of Internet use may serve as predictors of EDs.

We assumed those could be excessive BMI, communication with others as the main purpose of Internet use, and greater number of hours spent online on non-work/study-related activities (as a result of dissatisfaction with their body mass, people at risk of EDs are hypothesized to move their social lives online; in addition, individuals with EDs and PIU are reported to have a higher BMI due to, among others, lower physical activity [24]).

Due to the reported higher risk for both PIU [29,35] and EDs [36] in this very population, the study sample was composed of students. The added value of this research is the strict division of participants based on their selected major. To the best of our knowledge, very few studies to date [15,16,37,38] have taken into account the links between the field of study and the prevalence of comorbid PIU and EDs among students. Meanwhile, there exist reports indicating differences in PIU severity depending on the selected field of study [15,16]. These results indicate that students of some fields of study (e.g., students of arts or communication) are more likely to use the Internet more often. The field of study could therefore constitute a disturbing variable in the own study. Hence, in order to establish the scale of prevalence and comorbidity of PIU and EDs among students, the research should include representatives of various university majors. We decided to recruit first-cycle students, as they are on the verge of entering adulthood, often leaving their family home, enjoying new-found freedom—which puts them at particular risk of becoming addicted to the Internet [39].

Our results will contribute to the existing body of evidence on the prevalence and comorbidity of PIU and EDs as well as identify predictors of EDs among the characteristics of computer use, such as the purpose, manner and duration of online activity. This will enrich the current state of knowledge about the relationship between PIU and EDs.

## 2. Material and Methods

### 2.1. Study Procedure

This cross-sectional study was conducted in November and December 2018 at three universities of different profiles (medical, technical and economic), located in different regions of Poland (Gliwice, Katowice, Lublin). Upon informed consent, the students were asked to complete provided questionnaire sets by a trained interviewer in the lecture hall, after a class. Participation was voluntary with each of the participants informed about the study purpose and the possibility of withdrawing at any stage. Participants were not rewarded for participation. The study was approved by the Bioethics Committee of the Pomeranian Medical University in Szczecin (decision no: KB-0012/188/05/17).

### 2.2. Participants

A total of 1008 students of 1st cycle studies/years 1–3 of single-cycle programmes completed provided questionnaire sets. Such response rate allowed to obtain the statistical power of 1 (α = 0.05) in relation to H0, postulating no correlation between the results of the Problematic Internet Use Test and the Eating Attitudes Test, thus leaving a very low chance of type two error occurring.

The mean age of the participants was 21.3 years (range = 18–40, SD = 2.65). The sample included 510 women (50.6%) aged 18–40 and 498 men (49.4%) aged 19–39. There were no statistical differences in the number of male and female students of:(1)Medical and health sciences (336 students: 174 women and 162 men);(2)Humanities and social sciences (336 students: 168 women and 168 men);(3)Technical sciences (336 students: 168 women and 168 men).

Study Inclusion Criteria were:Enrollment in full-time, first-cycle studies or 1st, 2nd or 3rd year of single cycle studies (applicable to medical and dental degrees). Sample selection was dictated by the particular risk of Internet addiction reported in this population [39].Student status at a medical, economic or technical university. The study included an equal number of students of three groups: medical and health sciences; humanities and social sciences and technical sciences. Sample selection was dictated by the aim to compare the risk of PIU and EDs depending on the selected major, as available literature indicates the existence of such a relationship in terms of PIU [15,16].Sex of the participants. We intended to include an equal number of men and women at each university in order to make comparisons between the sexes in terms of the analyzed variables.The sample size was calculated at 663 using the EPI Info^TM^ 7.2.4.0 (the Centers for Disease Control and Prevention, Atlanta, Georgia, USA). The incidence of PIU was estimated at 10% based on available literature data [24,27]; the number of full-time undergraduate students in Poland was 895.725 [40], confidence level was 99.0%, confidence limits 5%.

### 2.3. Measures

#### 2.3.1. The Problematic Internet Use Test TPUI22 (PIU)

The research used the Polish-language, modified adaptation of the Internet Addiction Test (IAT by Kimberly Young), created by Ryszard Poprawa. This 22-item self-report uses a Likert-type scale ranging from 0 (“never”) to 5 (“always”). Participants can score from 0 to 110 points. Higher scores indicate greater problematic Internet use. Based on the obtained results, the following PIU categories can be identified: very low risk of Internet addiction (0–1 points for individuals of ≤24 years and 0 points for those of >24 years), low risk of Internet addiction (2–10 points for individuals of ≤24 years and 1–6 points for those of >24 years), moderate risk of Internet addiction (11–49 points for individuals of ≤24 years and 7–41 points for those of >24 years), high risk of Internet addiction (50–79 points for individuals of ≤24 years and 42–75 points for those of >24 years) and very high risk of Internet addiction (80–110 points for individuals of ≤24 years and 76–110 points for those of >24 years). The tool is reported to have good reliability (Cronbach’s alpha = 0.935) [41].

#### 2.3.2. The Eating Attitudes Test (EAT-26)

In addition, the research used the EAT-26 Test in the Polish adaptation by Włodarczyk-Bisagi et al. The questionnaire contains 26 questions with responses on a 5-point scale from “never” to “always”. The score range is from 0 to 78 points, with a cut-off of 20, indicating a risk of EDs. The tool has good reliability (Cronbach’s alpha = 0.84) [42,43]. It is the most commonly used tool to study the relationship of EDs with PIU [29].

#### 2.3.3. A Self-Designed Socio-Demographic and Internet Use Survey

The research made a self-designed survey called the Socio-demographic Variables and Internet Use Survey, and it included collecting information on sex of the participants, age, type of studies, extracurricular activity, self-reported financial status, amount of free time, as well as amount, manner and purposes of using the Internet.

### 2.4. Statistical Analysis

Statistical analysis was performed with the use of IBM SPSS Statistics v. 25 (IBM Corp., Armonk, NY, USA) using the Chi-square test (with Fisher’s exact test for 2 × 2 tables) and Student’s t-test to determine sex differences, Mann–Whitney U-Test to compare 2 groups of unequal sizes, pairwise Spearman’s rho for the assessment of relationships between continuous variables, one-way ANOVA with an additional Brown–Forsyth correction, and a linear stepwise regression for detected predictors of ED among females and males.

## 3. Results

### 3.1. Sociodemographic and Internet Use Characteristics in the Studied Group

As regards sociodemographic data, there were significant differences between women and men in terms of age (*p* < 0.001), self-reported financial status (compared to women, more men assessed their financial situation as very good or bad; *p* < 0.001), and extracurricular activity—more men worked (*p* = 0.007) and were involved with student organizations (*p* < 0.001) (Table 1).

As regards Internet use, women were more likely to go online using the computer (*p* < 0.001), to use the Internet to communicate with other people (*p* < 0.001) and use social media (*p* < 0.001), while men mostly used the Internet for academic (*p* < 0.001), work (*p* < 0.001) and entertainment purposes (*p* < 0.001). Women also used the Internet more than men for study and work-related purposes at weekends (*p* = 0.015). The average number of hours spent online on other activities (e.g., shopping or internet banking), both during working days and on weekends, was comparable for women and men (Table 1).

### 3.2. PIU and EAT-26 Scores in the Studied Group

A total of 103 (10.22%) of the respondents met the criteria of Internet addiction (“high” or “very high risk of PIU”). Interestingly, there was a significant sex difference, as the addicted group was represented by only 18 (3.5%) women compared to as many as 85 men (17.0%) [X2 (4) = 54.318; *p* < 0.001]. A total of 93 participants (9.23%), including 55 women (10.8%) and 38 men (7.6%) received above-cutoff EAT-26 scores (i.e., ≥20 points), which did not turn out to be a significant difference [X2 (1) = 2.992; *p* = 0.102].

We observed sex differences both in terms of PIU and EAT-26 scores (Table 2). Men received higher PIU scores (*p* < 0.001), while women obtained higher EAT-26 scores (*p* < 0.05).

### 3.3. Correlation between PIU and EAT-26, and EAT-26 Score Predictors

There were significant correlations between EAT-26 and PIU scores both in the entire sample (men and women together) (rho = 0.212; *p* < 0.001) and separately among women (rho = 0.219; *p* < 0.001) and men (rho = 0.232; *p* < 0.001).

Regression analysis of both the overall PIU score and its individual items demonstrated significant qualitative sex differences in terms of EAT predictors. Significant predictors of EAT-26 score in the entire sample were 3 PIU items and 3 additional variables [R = 0.368; R^2^ = 0.135; F (6) = 78.516; *p* < 0.001]. The strongest predictor was Item 14: “I feel preoccupied with the Internet”. Each additional point for this item resulted in an increase in the EAT-26 overall score by 0.194 points (*p* < 0.001). Each point for item 13 “I sometimes neglect sleep …” contributed to the increase of EAT-26 score by 0.115 points (*p* < 0.001), and an increase in Item 20 score: “online activity allows me to relieve my negative feelings”, which entailed an increase in the EAT-26 score by 0.106 points (*p* = 0.004). As regards Internet use characteristics, a significant predictor of the EAT-26 score was the average number of hours spent online on weekends for study/work-related purposes (β = 0.060; *p* = 0.045), while among anthropometric variables—lower height (β = −0.106; *p* < 0.001) and higher BMI (body mass index, β = 0.063; *p* = 0.034).

In turn, separate analyses in male and female subgroups demonstrated that among all the measured variables, EDs in women could be predicted by seven factors, and in men—by two, as presented in Table 3 below.

The fact that there were more significant ED predictors identified in women, while R^2^ coefficient was lower than in men suggests that EDs in women may be a much more complex phenomenon. Even though there is a significant match between both models, the variables relating to Internet use fail to explain a large proportion of the variance of EDs in young people. It is approximately 16% for women and 19% for men. Apart from the differences in the number of ED predictors in both subgroups, it is interesting that the overall PIU serves as the EAT-26 score predictor only in men, and for each one additional point in the PIU score there is slightly more than 0.5 point of increase in the EAT-26 score. In men, as in women, PIU Item No. 2 was a significant but negative predictor of EDs. BMI turned out to significantly affect the EAT-26 result only in women, and the predictive significance of height, which was reported for the entire sample, was not observed in either of the two subgroups.

Furthermore, there were weak correlations between EAT-26 score and the Internet use mainly for work-related purposes (rho = 0.074; *p* = 0.018), time spent online on study/work-related activities (rho = 0.079; *p* = 0.012), and online activities other than learning and work (rho = 0.097; *p* = 0.002), but only on working days.

Certain relationships were also observed with regard to the participants’ non-academic activity. Namely, those who apart from studying also worked (relative to their non-working counterparts; Z = 3.109; *p* = 0.002) had significantly higher EAT-26 results. Likewise, higher EAT-26 scores were also recorded in those involved in student organizations (compared to those non-involved; Z = −1.699; *p* = 0.089), while significantly lower EAT-26 scores were observed in students who did not report any additional activities (Z = −2.144; *p* = 0.032).

Analysis of variance demonstrated correlations between EAT-26 score and self-reported financial status, as presented in Table 4. However, taking into account the sex of the participants, it turned out that it persisted only in women and amounted to 3.13 points (*p* = 0.032; F (2) = 35.621; *p* = 0.035), suggesting that women with a good financial status manifested more severe eating problems, while in men, financial status did not seem to play a role.

No correlation was found between EAT-26 score and type of study [F (2) = 0.485; *p* = 0.616], main Internet use device [F (2) = 0.793; *p* = 0.454], age (rho = 0.011; *p* = 0.721) or year of study (rho = 0.050; *p* = 0.112).

## 4. Discussion

Globalization and technological progress bring numerous benefits and greatly facilitate everyday life. Nevertheless, they are not devoid of certain threats, such as, e.g., imposing global patterns of the human body or excessive digitization of life. It, therefore, seems critical to explore how these problems may affect various aspects of the lives of contemporary young adults—a generation growing up with full and majorly unrestricted access to the Internet, whose task in the near future will be to shape the next generation to come. The aim of this study was to investigate the prevalence of PIU and EDs, potential correlations between the two phenomena and ED predictors among sociodemographic and Internet use characteristics in Polish students. To this end, we surveyed equal-numbered groups of male and female students from three different universities, majoring in science/technology, humanities/social sciences and medical/health sciences.

### 4.1. The Prevalence of PIU and EDs

Our pooled results suggest a 10.22% incidence of PIU in the investigated sample (18 women and 85 men). Much in line with our expectations, PIU criteria were significantly more prevalent in men. Similar prevalence rates of IA (Internet addiction) were reported in Turkey (10.1%) [24] and France (10.2%) [27] and in Colombia (10%) [44]. Our results remain in accordance with those of other authors, who also found higher prevalence of IA in men compared to women [4,5,11,12,13,14,24]. This quite puzzling phenomenon can perhaps be accounted for by the purpose of online activity. Our results suggest that men use the Internet mainly for study, work and entertainment-related purposes. This may be congruous with the theory of specific addiction to the Internet, according to which one becomes addicted not to the Internet itself so much as to the activity performed online. In our sample, women, in turn, considered the Internet more of a means of communication with other people. Similar differences concerning the purpose of Internet use were also reported elsewhere [11,45,46]. Nevertheless, such dependence between sex of the participants and PIU does not seem to be a rule. In a big Chinese study, only 4.5% of the participants were deemed Internet dependent based on Young’s Internet addiction test, regardless of their sex [25]. Spanish and Mexican studies show that the differences between men and women in the PIU results concern its individual components [47,48]. The research conducted so far indicates the existence of the influence of culture on PIU. Researchers have shown that the Japanese achieve a higher PIU level than representatives of other ethnic groups [49,50]. In addition, residents of other parts of Asia score higher than representatives of European and African countries [5]. This supports the thesis about the cultural context of PIU. The reason for this variation may be the greater digitization of Asian countries.

In our study, we recorded an EAT-26 score of ≥20, considered indicative of high-risk state of EDs, in 9.23% of the investigated sample (55 women and 38 men). Contrary to expectations, we found no significant sex differences. However, analyses of raw scores still suggested their higher values in women compared to men. Higher EAT-26 rates (15.2%) were reported in Turkish youth (aged 14–20 years), with no significant sex differences [24]. Asian researchers report prevalence of EDs below 10% in Indonesia, Thailand and Vietnam, 13.8% in Malaysia and 20.6% in Myanmar, again with no differences between male and female populations [51]. Egyptian research on a considerable sample of 2.365 students seems to deserve particular attention, as the authors report a 17.9% incidence rate of EDs in the investigated population (more prevalent in women), and higher mean total EAT scores in women [37]. Such results may be attributed to women’s greater preoccupation with the appearance of their bodies [34]. Socio-cultural factors influence the development of eating disorders, especially the media and the promotion of thinness [31]. An example is the increase in the incidence of bulimia nervosa symptoms after exposure to the Western culture [30,52]. However, it is worth adding that other studies indicated the occurrence of anorexia nervosa also in populations not exposed to the influence of the Western culture, although with one difference—these people showed a lesser fear of gaining weight or becoming fat [30]. The influence of ethnicity on the occurrence of EDs has not been clearly assessed [30]. Some studies support the thesis that the occurrence of EDs is differentiated by ethnicity [42], and researchers indicate that culturally conditioned acceptance of larger body sizes is a factor protecting against the development of anorexia nervosa [53]. Hence, this problem is more common in countries influenced by the Western culture.

### 4.2. Correlation between PIU and EDs

Certain behaviors are associated with the production of a short-term reward. Repeating these behaviors may be persistent [54] and lead to impairment in personal, family, social, educational, occupational areas of functioning and distress [55]. Despite the lack of a psychoactive substance, the addictive mechanism has common foundations (i.e., salience, mood modification, tolerance, withdrawal, conflict and relapse) [56,57,58]. In the International Statistical Classification of Diseases and Health Problems ICD-11, one of the types of Internet addiction, which is “addiction to games”, was introduced in the subgroup “addictive behavior” [59,60,61]. The Internet use addiction category was included in the supplement to DSM-5 (Diagnostic and Statistical Manual of Mental Disorders) with the indication that it is a unit requiring further research [19].

Our initial hypothesis was that PIU scores would be higher in individuals with EDs. In line with our expectations, we find a link between PIU and EDs, both in men and women. Although the strength of the observed correlation was low (rho = 0.212), it still proved statistically significant (*p* < 0.001). The hypothesized correlation between Internet addiction and EDs was confirmed by Hinojo-Lucena et al. in their meta-analysis conducted in 2019 [29]. In a Turkish study carried out on a group of 584 high school students, a correlation was found between the Internet Addiction Test and the EAT-26 scores. In addition, it was observed that the presence of disordered eating attitudes, male sex, and high BMI were strong predictors of IA [24]. Significant links between PIU and EDs were also recorded in Egyptian students [37]. Likewise, the relationship was demonstrated in a group of 1199 Chinese secondary school and college students aged 12–25 years [25], as well as in French [27] or Asian populations [51]. At this point, a very important question arises: where does this correlation come from? Tao and Liu suggest that the unclear correlation of IA and EDs may reflect a parallel symptomatology rather than point to a causal nature of the relationship [25]. Other authors highlight the fact that the Internet, similar to so many other mass media (TV, fashion magazines), contributes to the promotion of a slim body image, hence its frequent users are likely to be dissatisfied with their physical appearance [32,37,62]. Or vice versa—the Internet is a place of refuge, with online activity a coping mechanism for those dissatisfied with their looks [27]. It should be added that the negative perception of a non-slim body image (not even overweight) is already widespread both in people with and without eating disorders [63], and the Internet may play here a role, if not a cause then an enhancer of eating disorders. The Internet could become one of the media, in addition to television, press and radio, influencing the promotion of a slim body shape. In addition, Internet users can communicate with each other using social media, where anyone can post various types of content. Some of this content promotes and encourages ED (e.g., groups that promote eating disorders as a lifestyle) [64,65]. People who use the Internet more often are therefore more exposed on its impact, which may, as research suggests, affect offline ED behavior [66]. It is pointed out that the coexistence of addictions and EDs may be accounted for by similar symptoms of both disorders (lack of control, craving, and denial) [36,51,67]. It has been suggested that addictive problems may also be part of the course of bulimia [68]. Our research seems to offer a tentative explanation underlying the observed correlation between PIU and EDs in the form of ED predictors found among characteristics of Internet use.

### 4.3. Predictors of Eating Disorders

We hypothesized that the predictors of EDs would be excessive BMI, Internet use with a primary aim to communicate with others, and spending a significant number of hours online on non-work/study related activity. Contrary to expectations, this hypothesis, apart from high BMI, has not been confirmed. What emerged was the image of Internet users not so much transferring their social lives to the web due to dissatisfaction with their body image, but rather people using the Internet to carry out tasks related to their study/work or other undertaken activities. A significant predictor of EDs turned out to be the average number of hours spent online on weekends for study/work-related purposes. The likelihood of eating problems increased as the scoring for the item—"I neglect my household chores in favor of longer on-line activity” decreased. Given the clinical manifestation and the psychological underpinnings of EDs, this may be related to the excessive perfectionism observed in this type of psychopathology. It is further supported by the negative prediction of the EAT-26 score for the item—"my achievements at work/school suffer due to my excessive online activity” observed in women. Participants who, apart from studying, were involved in other activities (worked or were active members of student organizations) reported significantly higher EAT-26 scores. Their results may, therefore, suggest the importance of striving for perfectionism, the need for external confirmation of competence or maintenance of control over their lives, often observed in the course of eating disorders, which may shed some new light on the relationship between PIU and EDs. These findings are consistent with reports by Casale et al., indicating that socially prescribed perfectionism may have an effect on problematic use of internet communication services [69]. In turn, in the studies by Rodgers et al., disordered eating patterns in women were positively correlated with time spent on social and communication websites [27]. In our study, higher EAT-26 scores were also predicted by lower height and higher BMI, i.e., physical features so diverse from the standard patterns of beauty imposed by the contemporary world. Alpaslan et al. propose that individuals with EDs caused by PIU may have higher body weight, e.g., due to a sedentary lifestyle [24]. Among the predictors of elevated EAT-26 scores we also found preoccupation with the Internet, neglect of sleep in favor of online activity, and relieving negative feelings while online. What is more, we observed higher severity of eating problems in women with self-reported high financial status, which remains in line with the findings of Pengpid et al., which suggested that wealthier students were at greater risk of developing EDs than their less affluent counterparts [51].

However, we found no links between the selected university major and EDs. Similarly, no such relationships were reported either by Kamal and Kamal in their research on students of various faculties (medical sciences, health sciences, arts, agriculture) [37] or Spillebout et al., although in the latter study the majority of students (53.5%) were “a mixed university group” [38]. In our sample, whose participants were recruited strictly according to their selected field of study, with an equal number of students in each group (science, humanities/social sciences, medical/health sciences), and a comparable number of men and women in each group, we did not observe any differences in ED prevalence across these subgroups.

In conclusion, this research indicates that there indeed exists a relationship between PIU and EDs and that predictors of EDs can be found among the characteristics of Internet use. Our results also suggest that individuals at risk of EDs use the Internet as a means to fulfill their study/work-related duties rather than to move their social interactions to virtual reality. Nevertheless, these findings require further confirmation. Therefore, more research is warranted to disambiguate this issue, especially with an aim to establish the direction of the relationship between PIU and EDs and the role of perfectionism in the development of PIU.

### 4.4. Limitations

This study is not exempt from a number of limitations. The first limitation results from the very nature of cross-sectional research. This type of research excludes the possibility of a clear determination of the cause-and-effect relationship between ED and PIU. Another noteworthy limitation is the small sample size, i.e., the number of problematic Internet users exceeding the cut-off score in the EAT-26 test. We did not take sexual orientation into account in the research, due to the fact that disclosing one’s sexual orientation is not common in Poland. Asking about one’s orientation could lead to withdrawal from the survey as a question that is too intrusive into one’s intimate sphere. However, we will consider this variable in subsequent studies, especially considering reports on the role of sexual orientation in eating concerns and body image problems in men [70]. Further research is needed to establish the causality of EDs and PIU.

## 5. Conclusions

There is a correlation between eating disorders and problematic internet use both in the entire sample and in the male and female subgroups.

Preoccupation with the Internet, neglect of sleep in favor of online activity, and relieving negative feelings while online seem to constitute significant predictors of eating disorders.

Other predictors of eating disorders include a higher average number of hours spent online on weekends on study/work-related activities, extracurricular activity (working, active membership in student organizations), lower height and higher BMI.

Further research is needed to establish the causality of the relationship between problematic Internet use and eating disorders and the effect of perfectionism on this relationship.

## Figures and Tables

**Table 1 ijerph-18-10269-t001:** Sociodemographic and Internet use characteristics including sex differences.

Variable		Entire Sample, *N* = 1008	Women, *n* = 510	Men, *n* = 498	T/X^2^	df	*p*
age; M (SD) median = 21.0		21.31 (2.65)	20.74 (2.36)	21.89 (2.80)	−7.103	1006	<0.001
Type of studie; *N* (%)	medical and health sciences	336 (33.3)	174 (34.1)	162 (32.5)	0.286	2	0.867
	humanities/social sciences	336 (33.3)	168 (32.9)	168 (33.7)			
	technical sciences	336 (33.3)	168 (32.9)	168 (33.7)			
Extracurricular activity; *N* (%)	work	515 (51.1)	239 (46.9)	276 (55.4)	7.387	1	0.007
	volunteer work	130 (12.9)	59 (11.6)	71 (14.3)	1.621	1	0.222
	care for a family member	93 (9.2)	45 (8.8)	48 (9.6)	0.200	1	0.665
	student organization activity	110 (10.9)	33 (6.5)	77 (15.5)	20.952	1	<0.001
	none of the above	391 (38.9)	219 (42.9)	172 (34.5)	7.493	1	0.007
Self-reported financial status; *N* (%)	very good	297 (29.5)	136 (26.7)	161 (32.3)	20.526	2	<0.001
	sufficient	662 (65.7)	362 (71.0)	300 (60.2)			
	bad	49 (4.8)	12 (2.4)	37 (7.4)			
Main Internet use device; *N* (%)	computer	108 (10.7)	30 (5.9)	78 (15.7)	67.220	2	<0.001
	smartphone/tablet	462 (45.8)	295 (57.8)	167 (33.5)			
	comparably computer and smartphone/tablet	438 (43.5)	185 (36.3)	253 (50.8)			
Main purpose of Internet use; *N* (%)	studying	284 (28.3)	105 (20.7)	179 (36.0)	39.955	5	<0.001
	work	66 (8.2)	29 (7.3)	37 (9.1)	40.434	5	<0.001
	entertainment	161 (16.0)	63 (12.4)	98 (19.7)	34.512	5	<0.001
	communication with other people	356 (35.5)	231 (45.5)	125 (25.2)	62.654	5	<0.001
	social media	128 (12.8)	77 (15.2)	51 (10.3)	48.235	5	<0.001
	other, e.g., shopping or internet banking	50 (5.0)	21 (4.2)	29 (5.9)	15.367	5	0.009
Average number of hours spent online on study/work-related activities; M (SD)	weekdays (Mon-Fri)	3.17 (5.06)	3.25 (3.71)	3.09 (6.15)	0.504	1006	0.614
	weekends	3.08 (2.06)	3.28 (2.66)	2.87 (1.61)	2.425	1006	0.015
Average number of hours spent online on other activities; M (SD)	weekdays (Mon-Fri)	2.95 (3.35)	2.86 (2.25)	3.97 (2.69)	−0.909	1006	0.363
	weekends	4.03 (3.70)	3.05 (4.19)	4.10 (4.50)	−0.560	1006	0.575

*N* = sample size; *n* = subsample size; M = mean; SD = standard deviation; T = Student’s *t*-test value; X^2^ = Pearson’s chi-square test value; df = degrees of freedom; *p* = significance.

**Table 2 ijerph-18-10269-t002:** PIU and EAT-26 scores in the entire sample, with male and female subgroups.

Test Type	Entire Sample, *N* = 1008		Women, *n* = 510		Men, *n* = 498		Student’s *t*-test		
	Min-Max.	M (SD)	Min-Max.	M (SD)	Min-Max.	M (SD)	*t*	df	*p*
PIU	0.00—110.00	25.84 (20.35)	0.00—97.00	21.83 (14.42)	0.00—110.00	29 95 (24.34)	−6.461	1006	<0.001
EAT	0.00—75.00	8.67 (9.48)	0.00—75.00	9.38 (9.69)	0.00—63.00	7.94 (9.21)	2.412	1006	0.016

PIU = problematic Internet use Test; EAT = Eating Attitudes Test; M = mean; SD = standard deviation; t = Student’s *t*-test value; df = degrees of freedom; *p* = significance.

**Table 3 ijerph-18-10269-t003:** Linear regression analysis of EAT-26 scores in men and women.

Subgroup	Predictors	*t*	*p*	Beta	F	df	*p*	R^2^
Women, *n* = 510	PIU item no 14 I feel preoccupied with the Internet when I am offline and I fantasize about being online	4.950	<0.001	0.236	12.985	7	<0.001	0.155
	PIU item no 13 I sometimes neglect sleep in favor of long online activity	4.261	<0.001	0.219				
	PIU item no 2 I neglect my household duties in favor of longer online activity	−2.260	0.024	−0.104				
	BMI	2.883	0.004	0.119				
	Average number of hours spent online at weekends for non-professional activities	−2.647	0.008	−0.114				
	PIU item no 11 I fear that my life without the Internet would be boring, empty and sad	2.237	0.026	0.104				
	PIU item no 15 My work/school achievements suffer due to my excessive online activity	−2.042	0.042	−0.105				
Men, *n* = 498	PIU total score	9482	<0.001	0.505	56.282	2	<0.001	0.186
	PIU item no 2 I neglect my household duties in favor of longer online activity	−2.506	0.013	−0.133				

PIU = problematic Internet use Test; EAT-26 = Eating Attitudes Test; t = Student’s *t*-test value; *p* = significance, F = variance value; df = degrees of freedom; R^2^ = total variance explained; BMI = body mass index.

**Table 4 ijerph-18-10269-t004:** Comparison of EAT severity depending on financial status.

Self-Reported Financial Status		Difference of Means	Significance	95% Confidence Interval		ANOVA		
				Lower Limit	Upper Limit	F	df	*p*
very good	sufficient	2.202	0.018	0.28	4.12	5.710	2	0.014
	bad	0.828	0.942	−3.17	4.83			
bad	sufficient	1.373	0.734	−5.04	2.29			

F = variance value; df = degrees of freedom; *p* = significance.

## Data Availability

The data presented in this study are available on request from the corresponding author.

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
