# Peer review of "There Are Predictors of Eating Disorders among Internet Use Characteristics—A Cross-Sectional Study on the Relationship between Problematic Internet Use and Eating Disorders"

_ijerph, 2021, doi:10.3390/ijerph181910269_

Round 1
Reviewer 1 Report
The manuscript is well written and fits a topic of high interest in psychiatry. The results are clearly presented. I suggest to the author's to implement the part on the "addictive behaviour" in the discussion section, paragraph "Correlation between PIU and EDs".
Author Response
Thank you very much for all Your comments - we have made the following changes before publication.
Point 1: ‘I suggest to the author's to implement the part on the "addictive behaviour" in the discussion section, paragraph "Correlation between PIU and EDs"’.
Response 1: We have added the following part, see page 9.
“Certain behaviors are associated with the production of a short-term reward. Repeating these behaviors may be persistent [56] and may lead to impairment in personal, family, social, educational, occupational areas of functioning and distress [57]. Despite the lack of a psychoactive substance, the addictive mechanism has common foundations (i.e. salience, mood modification, tolerance, withdrawal, conflict and relapse) [58–60]. In the International Statistical Classification of Diseases and Health Problems ICD-11, one of the types of Internet addiction, which is "addiction to games", was introduced in the subgroup "Addictive behavior" [61–63]. “The Internet use addiction” category was included in the supplement to DSM-5 (Diagnostic and Statistical Manual of Mental Disorders) with the indication that it is a unit requiring further research [19].”
Reviewer 2 Report
The manuscript is about the evaluation of the relationships between eating disorders and problematic internet use. The sample is adequate and the methodology applied is well described. Overall the paper is interesting for the field.
My concerns about the paper are:
- paragraph 3.1: you reported different p-values from the table, please verify them
- you used the term "gender", have you evaluated this aspect or rather the sexes of the participants?
- have you considered sexual orientations? Indeed, you found no specific differences between subgroups but there is robust literature about the role of sexual orientation in eating concerns and body image problems in men (see Meneguzzo et al., 2020 - 10.1007/s40519-020-01047-7)
- looking at eating disorders, the main construct of these psychiatric disorders is the presence of body image concerns. I understood that this construct is poorly evaluated by your questionnaires, but I think that this aspect could be discussed from your perspective. For example, is it possible that problematic internet use could be linked to the evaluation of seen bodies, that is a specific aspect of the body dissatisfaction in EDs? (see for the body judgment for example Behrens et al., 2020 - https://doi.org/10.1002/erv.2812)
- in the discussion you evaluate differences with other studies, is it possible that there is a cultural influence in the results (European countries vs Asian countries for example)?
Author Response
Thank you very much for all Your comments - we have made the following changes before publication.
Point 1: “paragraph 3.1: you reported different p-values from the table, please verify them”
Response 1: We have made changes to the text, see page 6.
Point 2: “you used the term "gender", have you evaluated this aspect or rather the sexes of the participants?”.
Response 2: No, we have not assessed the gender aspect. We have replaced the term "gender" in the text with "sex of the participants".
Point 3: “have you considered sexual orientations? Indeed, you found no specific differences between subgroups but there is robust literature about the role of sexual orientation in eating concerns and body image problems in men (see Meneguzzo et al., 2020 - 10.1007/s40519-020-01047-7)”.
Response 3: We have added the following part in the "Limitations", see page 11:
“We did not take sexual orientation into account in the research, due to the fact that disclosing one's sexual orientation is not common in Poland. Asking about one's orientation could lead to withdrawal from the survey as a question that is too intrusive into one's intimate sphere. However, we will consider this variable in subsequent studies, especially considering reports on the role of sexual orientation in eating concerns and body image problems in men [72].”
Point 4: “looking at eating disorders, the main construct of these psychiatric disorders is the presence of body image concerns. I understood that this construct is poorly evaluated by your questionnaires, but I think that this aspect could be discussed from your perspective. For example, is it possible that problematic internet use could be linked to the evaluation of seen bodies, that is a specific aspect of the body dissatisfaction in EDs? (see for the body judgment for example Behrens et al., 2020 - https://doi.org/10.1002/erv.2812)”.
Response 4: We added in the discussion ( part 3.2 Correlation between PIU and ED) :
“It should be added that the negative perception of a non-slim body image (not even overweight) is already widespread both in people with and without eating disorders [65], and the Internet may play here a role, if not a cause then an enhancer of eating disorders. The Internet could become one of the media, in addition to television, press and radio, influencing the promotion of a slim body shape. In addition, Internet users can communicate with each other using social media, where anyone can post various types of content. Some of this content promotes and encourages ED (e.g. groups that promote eating disorders as a lifestyle) [66,67]. People who use the Internet more often are therefore more exposed on its impact, which may, as research suggests, affect offline ED behavior [68].”
Point 5: “in the discussion you evaluate differences with other studies, is it possible that there is a cultural influence in the results (European countries vs Asian countries for example)? ”.
Response 5:
We added in the discussion, see page 9:
„The research conducted so far indicates the existence of the influence of culture on PIU. Researchers have shown that the Japanese achieve a higher PIU level than representatives of other ethnic groups [49,50]. Also, residents of other parts of Asia score higher than representatives of European and African countries [5]. This supports the thesis about the cultural context of PIU. The reason for this variation may be the greater digitization of Asian countries.”
“Socio-cultural factors influence the development of eating disorders, especially the media and the promotion of thinness [52]. An example is the increase in the incidence of bulimia nervosa symptoms after exposure to the Western culture [53,54]. However, it is worth adding that other studies indicated the occurrence of anorexia nervosa also in populations not exposed to the influence of the Western culture, although with one difference - these people showed a lesser fear of gaining weight or becoming fat [30]. The influence of ethnicity on the occurrence of EDs has not been clearly assessed [30]. Some studies support the thesis that the occurrence of EDs is differentiated by ethnicity [42], and researchers indicate that culturally conditioned acceptance of larger body sizes is a factor protecting against the development of anorexia nervosa [55]. Hence, this problem is more common in countries influenced by the Western culture. ”
Reviewer 3 Report
The article is relenta, but cuold be improved if the authors included some aspects as:
Sample: It would be interesting to explain the presence of students from the social area compared to those from science.
Discussion: In the The prevalence of PIU and EDs part, must included data from Spain, suggest cite: Marín-Díaz, V., & Sampedro, B.E. (2021). Social educators and their relationship with the Internet. Use or abuse of this medium. Digital Education Review, 39, 76-88. https://revistes.ub.edu/index.php/der/article/view/33190/pdf
and Peña, F., Rojas-Solís, J.L., & García-Sánchez, P-V. (2018). Uso problemático de internet, cyberculling y ciberviolencia de pareja en jóvenes universitarios. Diversitas: Perspectiva Sociológica, 14(2), 205-219. https://doi.org/10.15332/s1794-9998.2018.0002.01
and Colombia: Puerta-Cortés, D. X., & Carbonell, X. (2013). Uso problemático de Internet en una muestra de estudiantes
universitarios colombianos. Avances en Psicología Latinoamericana, 31(3). 620-631. https://revistas.urosario.edu.co/index.php/apl/article/view/223
The spanish author cuold be cite this article more cause the authors will be able to make comparisons.
Author Response
Thank you very much for all Your comments - we have made the following changes before publication.
Point 1: “Sample: It would be interesting to explain the presence of students from the social area compared to those from science.”.
Response 1:
We have added the justification for this group structure, see page 3:
„These results indicate that students of some fields of study (e.g. students of arts or communication) are more likely to use the Internet more often. The field of study could therefore constitute a disturbing variable in the own study”.
In the discussion, we commented on the result showing no differences between the EAT-26 score and the field of study (see page 10). Due to the lack of correlation between the EAT-26 result and the field of study, the paper does not present the result concerning the prevalence of PIU among students of particular fields of study. This result will be published in the next article on the prevalence of PIU among Polish students and its relationship with sociodemographic variables.
Point 2: “Discussion: In the The prevalence of PIU and EDs part, must included data from Spain, suggest cite: Marín-Díaz, V., & Sampedro, B.E. (2021). Social educators and their relationship with the Internet. Use or abuse of this medium. Digital Education Review, 39, 76-88. https://revistes.ub.edu/index.php/der/article/view/33190/pdf
and Peña, F., Rojas-Solís, J.L., & García-Sánchez, P-V. (2018). Uso problemático de internet, cyberculling y ciberviolencia de pareja en jóvenes universitarios. Diversitas: Perspectiva Sociológica, 14(2), 205-219. https://doi.org/10.15332/s1794-9998.2018.0002.01
and Colombia: Puerta-Cortés, D. X., & Carbonell, X. (2013). Uso problemático de Internet en una muestra de estudiantes universitarios colombianos. Avances en Psicología Latinoamericana, 31(3). 620-631. https://revistas.urosario.edu.co/index.php/apl/article/view/223
The spanish author cuold be cite this article more cause the authors will be able to make comparisons.”
Response 2:
We added in the discussion:
“Our pooled results suggest a 10.22% incidence of PIU in the investigated sample (18 women and 85 men). Much in line with our expectations, PIU criteria were significantly more prevalent in men. Similar prevalence rates of IA (Internet addiction) were reported in Turkey (10.1%) [24] and France (10.2%) [27] and in Colombia (10%) [44].”
“Spanish and Mexican studies show that the differences between men and women in the PIU results concern its individual components [47,48].”